# COVID-19-Associated Pulmonary Aspergillosis, Fungemia, and Pneumocystosis in the Intensive Care Unit: a Retrospective Multicenter Observational Cohort during the First French Pandemic Wave

Stéphane Bretagne,[a,b,c] Karine Sitbon,[a] Françoise Botterel,[d] Sarah Dellière,[a,b,c] Valérie Letscher-Bru,[e] Taieb Chouaki,[f] Anne-Pauline Bellanger,[g] Christine Bonnal,[h] Arnault Fekkar,[i] Florence Persat,[j] Damien Costa,[k] Nathalie Bourgeois,[l] Frédéric Dalle,[m] Florian Lussac-Sorton,[n] André Paugam,[c,o] Sophie Cassaing,[p] Lilia Hasseine,[q] Antoine Huguenin,[r] Nadia Guennouni,[s] Edith Mazars,[t] Solène Le Gal,[u] Milène Sasso,[v] Sophie Brun,[w] Lucile Cadot,[x] Carole Cassagne,[y] Estelle Cateau,[z] Jean-Pierre Gangneux,[aa] Maxime Moniot,[bb] Anne-Laure Roux,[cc] Céline Tournus,[dd] Nicole Desbois-Nogard,[ee] Alain Le Coustumier,[ff] Olivier Moquet,[gg] Alexandre Alanio,[a,b,c] Françoise Dromer,[a] on behalf of the French Mycoses Study Group

aInstitut Pasteur, Université de Paris, CNRS UMR2000, unité de Mycologie Moléculaire, Centre national de Référence Mycoses Invasives et Antifongiques, Paris, France

bLaboratoire de Parasitologie-Mycologie, Hôpital Saint Louis, Assistance Publique-Hôpitaux De Paris (AP-HP), Paris, France

cUniversité de Paris, Paris, France

dAssistance Publique-Hôpitaux De Paris (AP-HP), Hôpital Henri Mondor, Université Paris-Est Créteil Val-de-Marne, Créteil, France

eService de Parasitologie et de Mycologie Médicale, CHU de Strasbourg, Strasbourg, France

fLaboratoire de Parasitologie-Mycologie, CHU Amiens-Picardie, Amiens, France

gLaboratoire de Parasitologie-Mycologie, CHU Besançon, Besançon, France

hAssistance Publique-Hôpitaux De Paris (AP-HP), Laboratoire de Parasitologie-Mycologie, Hôpital Universitaire Bichat, Paris, France

iAssistance Publique-Hôpitaux De Paris (AP-HP), Groupe Hospitalier La Pitié-Salpêtrière, Service de Parasitologie Mycologie, Sorbonne Université, Inserm, CNRS, Centre d'Immunologie et des Maladies Infectieuses (CIMI), Paris, France

jHospices Civils de Lyon, Service de Parasitologie et Mycologie Médicale, Hôpital de la Croix-Rousse, Lyon–Université Claude Bernard Lyon 1, Lyon, France

kLaboratoire de Parasitologie-Mycologie, CHU Charles-Nicolle, Rouen, France

lLaboratoire de Parasitologie-Mycologie, CHU de Montpellier, Montpellier, France

mLaboratoire de Parasitologie Mycologie, Centre Hospitalier Universitaire de Dijon—Hôpital François Mitterrand, Dijon, France

nDepartment of Parasitology, Bordeaux University Hospital, Bordeaux, France

oAssistance Publique-Hôpitaux De Paris (AP-HP), Hôpital Cochin, Paris, France

pService de Parasitologie-Mycologie, Hôpital Purpan Toulouse, CHU Toulouse, Toulouse, France

qLaboratoire de Parasitologie Mycologie CHU de Nice, Nice, France

rParasitologie Mycologie-Laboratoire de Parasitologie-Mycologie, Pôle de Biopathologie, CHU de Reims, Université de Reims Champagne Ardenne, Reims, France

sAssistance Publique-Hôpitaux De Paris (AP-HP), Service de Bactériologie, Virologie, Parasitologie et Hygiène, Hôpital Necker-Enfants Malades, IHU Imagine, Paris, France

tCH de Valenciennes, Laboratoire de Microbiologie, Valenciennes, France

uLaboratoire de Parasitologie et Mycologie, Hôpital de La Cavale Blanche, CHU de Brest, Brest, France

vLaboratoire de Parasitologie Mycologie CHU Nîmes, Nîmes, France

wAssistance Publique-Hôpitaux De Paris (AP-HP), Laboratoire de Parasitologie Mycologie Hôpital Avicenne, Bobigny, France

xDépartement d'Hygiène Hospitalière, CHU Montpellier, Montpellier, France

yIHU Marseille—Institut Hospitalier Universitaire Méditerranée Infection, Marseille, France

zLaboratoire de Parasitologie-Mycologie, CHU de Poitiers, Poitiers, France

aaCHU de Rennes, Université de Rennes, Institut de Recherche en Santé, Environnement et Travail (IRSET), Rennes, France

bbLaboratoire de Parasitologie-Mycologie, CHU Clermont-Ferrand, Clermont-Ferrand, France

ccAssistance Publique-Hôpitaux De Paris (AP-HP), Hôpital Raymond Poincaré Garches, Hôpital Ambroise Paré, Boulogne Billancourt, France

ddLaboratoire de Microbiologie, Centre Hospitalier de Saint-Denis, Saint-Denis, France

**Citation** Bretagne S, Sitbon K, Botterel F, Dellière S, Letscher-Bru V, Chouaki T, Bellanger A-P, Bonnal C, Fekkar A, Persat F, Costa D, Bourgeois N, Dalle F, Lussac-Sorton F, Paugam A, Cassaing S, Hasseine L, Huguenin A, Guennouni N, Mazars E, Le Gal S, Sasso M, Brun S, Cadot L, Cassagne C, Cateau E, Gangneux J-P, Moniot M, Roux A-L, Tournus C, Desbois-Nogard N, Le Coustumier A, Moquet O, Alanio A, Dromer F, on behalf of the French Mycoses Study Group. 2021. COVID-19-associated pulmonary aspergillosis, fungemia, and pneumocystosis in the intensive care unit: a retrospective multicenter observational cohort during the first French pandemic wave. Microbiol Spectr 9:e01138-21. https://doi.org/10.1128/Spectrum.01138-21.

Address correspondence to Alexandre Alanio, alexandre.alanio@pasteur.fr.

[ee]Laboratoire de Parasitologie-Mycologie, Centre Hospitalier Universitaire de Martinique, Fort-de-France, La Martinique, France

[ff]Centre Hospitalier Intercommunal de Bigorre, Tarbes, France

[gg]Laboratoire de Parasitologie-Mycologie, Centre Hospitalier de Beauvais, Beauvais, France

**ABSTRACT** The aim of this study was to evaluate diagnostic means, host factors, delay of occurrence, and outcome of patients with COVID-19 pneumonia and fungal coinfections in the intensive care unit (ICU). From 1 February to 31 May 2020, we anonymously recorded COVID-19-associated pulmonary aspergillosis (CAPA), fungemia (CA-fungemia), and pneumocystosis (CA-PCP) from 36 centers, including results on fungal biomarkers in respiratory specimens and serum. We collected data from 154 episodes of CAPA, 81 of CA-fungemia, 17 of CA-PCP, and 5 of other mold infections from 244 patients (male/female [M/F] ratio = 3.5; mean age, $64.7 \pm 10.8$ years). CA-PCP occurred first after ICU admission (median, 1 day; interquartile range [IQR], 0 to 3 days), followed by CAPA (9 days; IQR, 5 to 13 days), and then CA-fungemia (16 days; IQR, 12 to 23 days) ($P < 10^{-4}$). For CAPA, the presence of several mycological criteria was associated with death ($P < 10^{-4}$). Serum galactomannan was rarely positive (<20%). The mortality rates were 76.7% (23/30) in patients with host factors for invasive fungal disease, 45.2% (14/31) in those with a preexisting pulmonary condition, and 36.6% (34/93) in the remaining patients ($P = 0.001$). Antimold treatment did not alter prognosis ($P = 0.370$). *Candida albicans* was responsible for 59.3% of CA-fungemias, with a global mortality of 45.7%. For CA-PCP, 58.8% of the episodes occurred in patients with known host factors of PCP, and the mortality rate was 29.5%. CAPA may be in part hospital acquired and could benefit from antifungal prescription at the first positive biomarker result. CA-fungemia appeared linked to ICU stay without COVID-19 specificity, while CA-PCP may not really be a concern in the ICU. Improved diagnostic strategy for fungal markers in ICU patients with COVID-19 should support these hypotheses.

**IMPORTANCE** To diagnose fungal coinfections in patients with COVID-19 in the intensive care unit, it is necessary to implement the correct treatment and to prevent them if possible. For COVID-19-associated pulmonary aspergillosis (CAPA), respiratory specimens remain the best approach since serum biomarkers are rarely positive. Timing of occurrence suggests that CAPA could be hospital acquired. The associated mortality varies from 36.6% to 76.7% when no host factors or host factors of invasive fungal diseases are present, respectively. Fungemias occurred after 2 weeks in ICUs and are associated with a mortality rate of 45.7%. *Candida albicans* is the first yeast species recovered, with no specificity linked to COVID-19. Pneumocystosis was mainly found in patients with known immunodepression. The diagnosis occurred at the entry in ICUs and not afterwards, suggesting that if *Pneumocystis jirovecii* plays a role, it is upstream of the hospitalization in the ICU.

**KEYWORDS** COVID-19, *Aspergillus*, fungemia, pneumocystosis, critical care, France, aspergillosis

The COVID-19 epidemic monopolizes the attention of health care services, with a lot of effort made to look for the best treatment and vaccination. Several publications reported fungal coinfections occurring during severe acute respiratory syndrome coronavirus 2 (SARS-CoV-2), with focus on COVID-19-associated pulmonary aspergillosis (CAPA), as invasive pulmonary aspergillosis complicates influenza in the intensive care unit (ICU) (1, 2). Invasive yeast infections are also reported associated with COVID-19 (3). Uncovering a specific pattern could help in designing preventive measures. Indeed, the possible immunodepression linked to COVID-19 and its treatment could increase the risk for yeast infection, in addition to the risks related to the ICU itself (4). The COVID-associated immunodepression could also stimulate the development of *Pneumocystis jirovecii*, responsible for pneumocystosis (PCP) in immunocompromised patients (5).

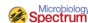

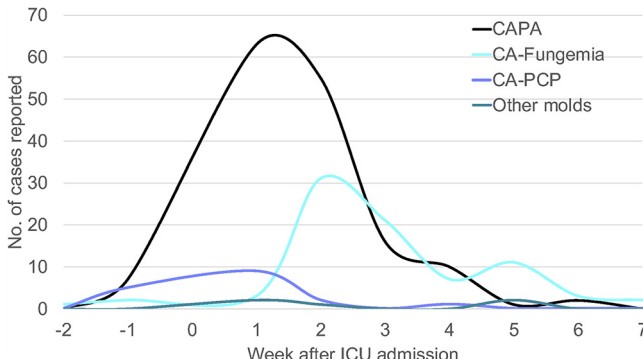

**FIG 1** Delays between admission in the ICU and diagnosis of fungal diseases of the 257 episodes of the present study with significantly difference according to fungal disease: CA-PCP occurred first ($n$ = 17; median delay, 1 day; interquartile range [IQR], 0 to 3 days), followed by CAPA ($n$ = 154; 9 days; IQR, 5 to 13 days) and CA-non-*Aspergillus* diseases ($n$ = 5; 11 days, IQR, 7 to 33 days), and CA-fungemia (16 days; IQR, 12 to 23 days) ($P < 10^{-4}$).

The French National Reference Center for Invasive Mycoses and Antifungals (NRCMA) launched a nationwide survey during the first pandemic wave to describe invasive fungal diseases and PCP associated with COVID-19.

## RESULTS

We collected data from 283 severe fungal infections, excluding 8 cases for lack of a COVID-19 PCR result, 17 that occurred outside the ICU (6 of CAPA, 4 of COVID-19-associated [CA]-fungemia, 6 of CA-PCP, and 1 of mucormycosis), and 1 episode of CAPA that lacked valid microbiological criteria. Therefore, we analyzed 257 episodes from 244 patients, with two different fungal infections in 13 patients (see Table S1 in the supplemental material). The median delay between the onset of the first symptoms and hospitalization in the ICU was 7 days (interquartile range [IQR], 4.75 to 10.25 days), and the delay between the COVID-19 PCR and hospitalization in the ICU was 0 day (IQR, −1 to 2 days).

There were 54 women and 190 men, with a mean age of 64.7 ± 10.8 years. The global mortality rate was 43.9% (107/244), and the rates were similar in both genders ($P$ = 0.317). The patients who died were older (70 ± 16 years) than those who survived (65 ± 13 years) ($P$ = 0.005). When restricted to the same period in the 12 Parisian ICUs, the global mortality rate for COVID-19 patients with fungal disease was 50.6% (45/89), compared to 22.6% (494/2,187) for all COVID-19 patients in the ICU during the same period ($P < 10^{-8}$).

There were 154 episodes of CAPA, 81 of CA-fungemia, 17 of CA-PCP, and 5 of COVID-associated non-*Aspergillus* infections (CA-non-*Aspergillus* infections) (Table S1), including one fusariosis (6). Delays between admission in the ICU and diagnosis significantly differed according to fungal disease (Fig. 1). CA-PCP occurred first (median delay, 1 day; IQR, 0 to 3 days), followed by CAPA (median, 9 days; IQR, 5 to 13 days) and CA-non *Aspergillus* diseases (median, 11 days; IQR, 7 to 33 days), with CA-fungemia as the latest diagnosis (median, 16 days; IQR, 12 to 23 days) ($P < 10^{-4}$).

**CAPA episodes.** For the 154 CAPA episodes, there was a male predominance (male/female [M/F] ratio = 3.5), with a mean age of 66 ± 9.7 years, and for the large majority (151/154), the episode was the first fungal disease. The global mortality was 46.1% (71/154). When comparing patients who died and those who survived, the mean age (67 ± 9.5 versus 65 ± 9.9 years, respectively; $P$ = 0.18) and the median times between ICU admission and CAPA diagnosis (9 days [IQR, 5 to 13 days] versus 8 days [IQR, 5 to 13 days]; $P$ = 0.608) did not differ significantly. No biopsy was reported, ruling out the categorization as proven cases. The two main mycological investigations performed were culture (153/154 [99.4%]) and serum galactomannan (GM) (132/153 [86.3%]) (85.7%) (Table 1). The diagnosis relied mainly on positive culture (76.5% [117/153]) of respiratory specimens, including

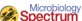

**TABLE 1** Main features of the 154 episodes of CAPA according to (i) the presence of host factors following EORTC/MSGERC definitions (group 1), (ii) the presence of a history of respiratory disease (group 2), or (iii) neither of these two conditions (group 3)

| Parameter[a] | Result for: | | | |
| --- | --- | --- | --- | --- |
| | Group 1 (n = 30) | Group 2 (n = 31) | Group 3 (n = 93) | P value |
| M/F ratio | 22/30 | 21/31 | 77/93 | 0.172 |
| Median age, yr (IQR = 25–75 yr) | 68 (59–71) | 68 (61–74) | 68 (60–73) | 0.590 |
| Mortality rate, no. (%) | 23/30 (76.7) | 14/31 (45.2) | 34/93 (36.6) | **0.001**[b] |
| Median days between hospitalization and diagnosis, no. (IQR = 25–75 days) | 9 (5–13) | 9 (4–13) | 9 (5–13) | 0.665 |
| Steroids for COVID-19, no. (%) (n = 146) | 11/28 (39.3) | 7/28 (25) | 31/90 (34.4) | 0.506 |
| Prescription of any antifungal drug for ≥3 days, no. (%) (n = 148) | 22/27 (78.9) | 25/31 (80.6) | 71/90 (78.9) | 0.948 |
| BAL fluid as respiratory specimens, no. (%) (n = 77) | 14/29 (48.3) | 15/30 (50) | 48/93 (51.6) | 0.928 |
| | | | | |
| Diagnostic means | | | | |
| Culture, no. (%) | | | | |
| Done (n = 153 [99.4%]) | 29 (96.7) | 31 (100) | 93 (100) | |
| Positive (n = 116 [76.8%]) | 25/29 (86.2) | 28/31 (90.3) | 63/93 (67.7) | **0.014**[b] |
| Direct examination, no. (%) | | | | |
| Done (n = 100 [64.94%]) | 21 (70.0) | 16 (51.6) | 63 (67.7) | 0.215 |
| Positive (n = 32 [32.0%]) | 7/21 (33.3) | 9/16 (56.3) | 16/63 (25.4) | 0.061 |
| BAL GM index of >1, no. (%) | | | | |
| Done (n = 59 [38.3%]) | 13/30 (43.3) | 5/31 (16.13) | 41/93 (44.1) | **0.018**[b] |
| GM index of >1 (n = 38 [64.4%]) | 6/13 (46.2) | 4/5 (80.0) | 28/41 (68.3) | 0.261 |
| BAL *Aspergillus* PCR, no. (%) | | | | |
| Done (n = 104 [67.5%]) | 23/30 (76.7) | 19/31 (61.3) | 62/93 (66.7) | 0.422 |
| Positive (n = 89 [85.6%]) | 20/23 (87.0) | 17/19 (89.5) | 52/62 (83.9) | 0.812 |
| Serum GM index of >0.5, no. (%) | | | | |
| Done (n = 132 [85.7%]) | 28/30 (93.3) | 23/31 (74.2) | 81/93 (87.1) | 0.085 |
| Positive (n = 25 [18.9%]) | 7/28 (25.0) | 2/23 (8.7) | 16/81 (19.8) | 0.320 |
| Serum *Aspergillus* PCR, no. (%) | | | | |
| Done (n = 61 [39.6%]) | 13/30 (43.33) | 9/31 (29.03) | 39/93 (41.93) | 0.400 |
| Positive (n = 11 [18.0%]) | 4/13 (30.76) | 1/9 (11.11) | 6/39 (15.38) | 0.236 |

[a]M/F ratio, male/female ratio; IQR, interquartile range; BAL, bronchoalveolar lavage; GM, galactomannan.
[b]$P < 0.05$ are highlighted in bold.

bronchoalveolar lavage (BAL) fluid (77/153 [50.3%]), endotracheal aspiration (74/153 [48.4%]), and sputum (2/153 [1.3%]).

Forty-nine patients received steroids (information available for 146 episodes), and the mortality rate was higher (31/49 [63.3%]) than that for patients who did not receive steroids (36/97 [37.1%]) ($P = 0.003$).

Prescription of antifungal drugs in 77.3% (119/154) of the patients included mostly voriconazole (85/119; 71.4%), but also liposomal amphotericin B (17/119; 14.3%), isavuconazole (8/119; 6.7%), caspofungin (7/119; 5.9%), and liposomal amphotericin B in association with either caspofungin or isavuconazole, with one case each. When excluding 7 patients who died <3 days after diagnosis, the mortality rate in the 30 patients who were not given antifungals (11/30 [36.7%]) was not significantly different from that of the 118 treated patients (54/118 [45.8%]) ($P = 0.370$). The global lengths in the ICU were 26 days (IQR, 16 to 36 days) and 28.5 days (20 to 40.25 days) (7 missing data) for the patients discharged alive.

The patients were grouped as follows: (i) the presence of immunosuppression factors following EORTC/MSGERC definitions (7), group 1; (ii) the presence of a history of respiratory disease (chronic obstructive pulmonary disease [COPD] or severe asthma), group 2; or (iii) neither of these two conditions, group 3. Groups 1, 2 and 3 comprised, respectively, 30, 31, and 93 episodes (Table 1) with statistically different mortality rates (76.7% [23/30], 45.2% [14/31], and 36.6% [34/93], respectively; $P = 0.001$), whereas the gender, the type of respiratory specimens, the administration of steroids, or the prescription of antifungal treatment/the antifungal drug prescribed had no impact on the mortality rate (Table 1). The only diagnostic means that differed among the groups was galactomannan in bronchoalveolar lavage fluid (BAL GM), which was performed

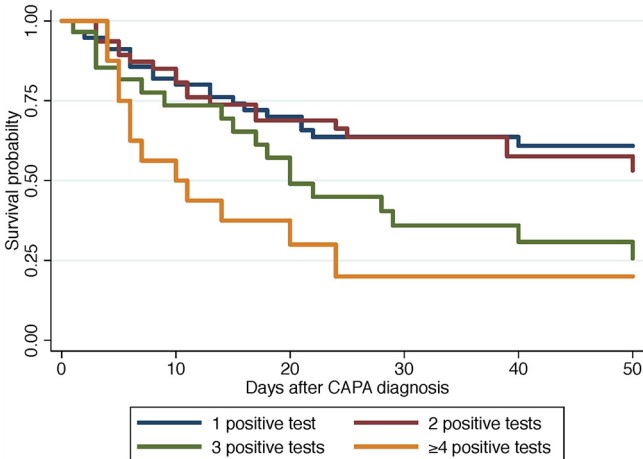

**FIG 2** Overall survival after diagnosis of CAPA according to the diagnostic mycological criteria with either one (blue line), two (brown line), three (green line), or more than 4 (yellow line) criteria in the whole population studied (Kaplan-Meier analysis, log rank test, $P = 0.0020$).

less often in group 2 (16.1%) than in the other groups (70 and 67.7%, respectively) ($P = 0.018$).

The positivity rates on respiratory specimens were as follows in decreasing order: PCR, 89/104 (85.6%); culture, 117/152 (76.5%); BAL GM $\geq$ 1, 38/59 (64.4%), and direct examination (32/100 [32.0%]). There was no difference between groups, except for culture more often being positive in groups 1 and 2 (89.7% and 90.3%, respectively) than in group 3 (67.7%) ($P = 0.012$) (Table 1). In contrast, serum markers were much less contributive, with GM $\geq$ 0.5 in only 18.9% (25/132) and PCR in 18.0% (11/61) of serum samples tested (Table 1). Only two episodes were considered CAPA on a positive serum *Aspergillus* PCR only. Overall, all patients in group 1 had probable invasive aspergillosis according to the EORTC/MSGERC definitions (7), while 123/124 patients (99.2%) had probable CAPA according to the recent proposed definition (8). Only one patient could have been classified as showing colonization (deceased 4 days after a positive *Aspergillus* PCR on sputum).

Cultures yielded mainly *Aspergillus fumigatus* (110/117 [94.0%]) but also *Aspergillus flavus* ($n = 2$ in group 1 [1.7%]), *Terrei* section ($n = 1$ in group 3 [0.9%]), *Nigri* section ($n = 2$ in group 1 and group 3 [3.4%]). Azole resistance was reported only once (9). *Aspergillus* quantitative PCR (qPCR) was performed in 72 (65.5%) of the 110 respiratory specimens that grew *A. fumigatus* and was positive in 64 (88.9%). Twelve culture-positive samples were PCR negative, of which four grew non-*fumigatus* species.

No single mycological criterion was associated with death (culture, $P = 0.345$; direct examination, $P = 0.250$; BAL GM $>$ 1, $P = 0.207$; serum GM, $P = 0.085$; serum *Aspergillus* qPCR, $P = 0.065$), except for respiratory *Aspergillus* qPCR ($P = 0.042$). When combined, the higher the number of positive microbiological criteria, the higher the mortality rate (log rank test, $P = 0.0020$) (Fig. 2). A similar trend was observed with patients of group 3, but without reaching statistical significance (log rank test, $P = 0.0768$) (Fig. 3).

**CA-fungemia episodes.** We analyzed 81 episodes, of which 6 occurred after a first episode of fungal infection (mainly CAPA) (Table S1). The patients (M/F ratio = 3.5) were significantly younger (65 $\pm$ 18 years) than patients with CAPA ($P = 0.037$). Mortality was 45.7% (37/81), with a median length of ICU hospitalization of 33 days (IQR, 18.35 to 51.75 days). When comparing patients who died and those who survived, the mean ages were significantly different (66.2 $\pm$ 10.4 years versus 60.4 $\pm$ 11.6 years, respectively; $P = 0.030$), as were the median delays between ICU hospitalization and diagnosis of fungemia (14 days [IQR, 10 to 17 days] versus 19.5 days [IQR, 12 to 30 days], respectively; $P = 0.0155$), while gender had no influence ($P = 0.676$). When considering the patients for whom information on steroid prescription was available, the mortality

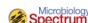

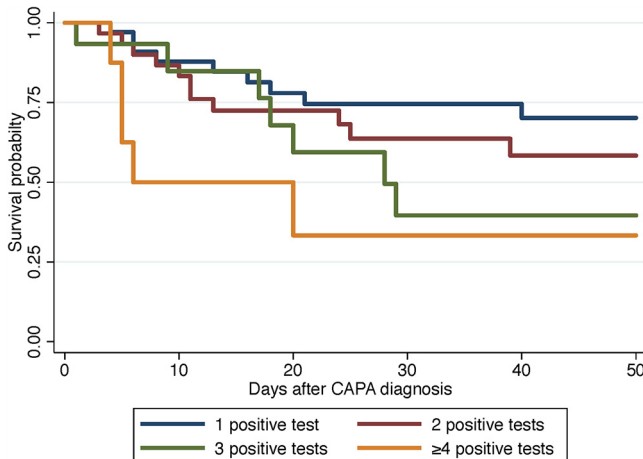

**FIG 3** Overall survival after diagnosis of CAPA according to the diagnostic mycological criteria with either one (blue line), two (brown line), three (green line), or more than 4 (yellow line) criteria in patients with no known host factors for invasive fungal diseases and with no history of severe respiratory disease (Kaplan-Meier analysis, log rank test, $P = 0.0768$).

rate was not significantly higher in those given steroids (18/32 [56.3%]) compared to those who were not (17/44 [38.6%]) ($P = 0.128$).

Information on antifungal treatment was provided for 80 patients, but five patients died within 48 h following diagnosis. The antifungal treatment for the 75 remaining patients was mainly ($n = 63$ [84%]) echinocandins (caspofungin, $n = 57$; micafungin, $n = 6$), but also included azoles ($n = 10$ [13.3%]), including fluconazole ($n = 7$), voriconazole ($n = 2$), and isavuconazole ($n = 1$ with concomitant CAPA), and liposomal amphotericin B ($n = 2$).

Eight yeast species were identified associated with different outcome (see Table S2 in the supplemental material). *Candida albicans* was the most frequent species ($n = 48$ [59.3%]). Of note, *Candida parapsilosis* and *Clavispora lusitaniae* CA-fungemias were associated with the lowest mortality rate and tended to occur the latest (Table S2). There was no significant difference in terms of age ($P = 0.9411$), mortality rate ($P = 0.780$), and prescription of steroids ($P = 0.776$) between group 1 ($n = 15$), group 2 ($n = 4$), and group 3 ($n = 62$).

**CA-PCP episodes.** Nine centers reported 17 patients (M/F ratio = 3.25; mean age, $58.2 \pm 14.5$ years; 1 to 5 episodes/center) had CA-PCP. Five patients died (29.4%); two of them had another concomitant infection (1 case of CA-fungemia and 1 of CAPA) (Table S1). The median stay in the ICU was 16 days (IQR, 8 to 20 days). The patients with CA-PCP mostly belonged to group 1 ($n = 10$) and group 3 ($n = 6$), and one was observed in group 2. Quantification cycle ($C_q$) values were obtained on different respiratory specimen types and thus not comparable, but fungal load was lower in groups 2 and 3. Serum $\beta$-D-glucan (BDG) testing was performed for eight patients, and only three were positive.

## DISCUSSION

This observational study of 257 episodes of COVID-associated fungal diseases that occurred during the first pandemic wave in France provides some insight into the occurrence of these infections. CAPA was the major infection (59.9%), in contrast to what is usually observed in the ICU, where fungemia is predominant (57.0% of the 3,374 cases of invasive mycoses recorded in ICUs over 7 years in our surveillance network, NRCMA [unpublished data]). The mortality rate was high (43.9%) and significantly higher than that recorded all causes combined in a subset of 12 ICUs (50.6% versus 22.6%; $P < 10^{-8}$). These figures are close to the published 52.8% mortality rate for CAPA and yeast infections combined (3), whereas the mortality is only 25.7% in the control group of the RECOVERY Study (10). Therefore, the occurrence of fungal

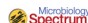

infection is associated with an increased mortality, but this is strongly dependent on the type of fungal diseases and underlying conditions.

CAPA here had a peak of occurrence at 9 days after ICU admission, comparable to those of previous publications (9 [11] or 8 [3] days) but longer than those in others (6 [12], 5.5 [13], 4 [14], or 2 to 8 days [15]). Numerous parameters can explain these 4 to 5 days in difference, including what is considered the day of diagnosis (onset of pulmonary symptoms, positive microbiological result, or initiation of specific therapy). Nonetheless, this relatively late diagnosis contrasts with influenza-associated acute pulmonary aspergillosis, the median onset of which is 3 days (1, 2). This peak at 9 days is compatible with an intrahospital acquisition of CAPA. Nothing is known about the presence of fungal spores and more specifically of *Aspergillus* spp. in ICUs, in contrast with hematology wards (16).

The second major observation for CAPA was that the prognosis worsened with the number of positive microbiological criteria globally but also in patients with no known risk factors for invasive aspergillosis or colonization (Fig. 2 and 3). This probably corresponds to a more advanced infection, as described in EORTC/MSG patients (17), and may explain why antimold drug prescription did not improve the prognosis, as was also reported in a recent systematic review (11). It is likely that clinicians delayed prescription of azoles until diagnosis was ascertained, knowing the possible interferences/side effects of this class of antifungals and the lack of demonstrated benefit (18). Since the prevalence of CAPA increases with deterioration of respiratory conditions (5.7 to 19.4% in reference 12), the sooner the prescription based on the first positive test, the better (8). Prophylaxis could be discussed as for pulmonary aspergillosis in critically ill influenza patients (18), although no benefit was observed in critically ill influenza patients (19). For the patients with EORTC/MSGERC host factors and COVID-19, more than 75% of them died, which is worse than the 58.4% in ICU patients with hematological malignancies and invasive pulmonary aspergillosis (20). Antifungal prophylaxis could be proposed for patients with EORTC/MSGERC host factors and COVID-19 as after allogeneic stem cell transplantation or intensive chemotherapy (21).

For CAPA occurring in non-EORTC/MSGERC patients, the first hurdle was CAPA definition. No criteria existed during the first COVID-19 wave, and the definition deeply impacts the analyses (3). Recently, Koehler et al. have proposed to differentiate between proven, probable, and possible CAPA following a format defined for EORTC/MSGERC patients (8). Differentiation between probable and possible CAPA relies on the respiratory specimens considered: from BAL for probable and nonbronchoscopic lavage for possible. However, the distinction between BAL, nondirected BAL, and nonbronchoscopic lavage arises from the difficulties in assessing the accountability of *Aspergillus* detection in respiratory specimens, with the assumption that culture from the deepest parts of the lung is more reliable/significant than culture from the trachea. However, choice of respiratory specimens is too dependent on local practices and the patient's condition. Furthermore, some authors recommend limiting BAL procedures based on the risk of SARS-Cov-2 aerosolization (18, 22). For us as for others, a positive endotracheal culture is sufficient to delineate probable aspergillosis for patient management (23, 24).

Respiratory specimens are all the more important as the blood biomarkers are less efficient. Serum GM and *Aspergillus* qPCR were rarely positive (<20%), as reported previously (13, 25). This lack of performance was not related to a lack of prescription for serum GM, which was tested in 85.7% of patients. This low yield may be explained by the rarity of neutropenia in COVID-19 patients, the main factor associated with serum GM positivity (26). This observation advocates for a lower invasiveness of the fungus during COVID-19. Therefore, if immunology and pathology studies suggest that SARS-CoV-2 could create an environment even more favorable for mold infection than influenza (27), the most deleterious effects may remain at the level of the bronchi without tissue invasion. Of note, no histologically proven case was reported here. A systematic review of autopsies of 677 decedents with COVID-19 has recently observed a 2% rate

of autopsy-proven CAPA (28). Thus, the association between mycological markers and mortality might not be linked to the invasiveness of the fungus.

The place of qPCR in respiratory specimens can be discussed. In contrast to serum, for which PCR results are now included in the EORTC/MSGERC criteria, PCR in respiratory specimens depends on so many variables that the decision is still pending (7). Hence our reluctance to define a threshold of positivity for the qPCR as proposed elsewhere (8). PCR assays also differed among centers, which makes drawing definitive conclusions difficult, even if some rules can be followed (29). Nevertheless, qPCR was widely used on respiratory specimens (67.5% of the cases). Since only 12 out of 72 (16.7%) culture-positive samples tested were qPCR negative, qPCR may appear redundant. Alternatively, qPCR may well replace culture when dealing with potentially hazardous samples. This hazard may explain why other means (direct examination or BAL GM) were less frequently performed because of the need for additional handling. Thus, culture was performed in almost all cases and was the most frequent criterion (76.3%). The species identified was mainly *A. fumigatus*. Of note, other mold species, such as *Mucorales*, were anecdotal, in contrast to other settings (30). Azole resistance in *A. fumigatus* was mentioned only once in the present study and published separately (9).

We show here a higher mortality in CAPA patients receiving steroid therapy, whatever the group, as already observed in CAPA (11, 13), in COVID-19-associated rhino-orbital-cerebral (30) and pulmonary (31) mucormycosis, and in bacterial coinfections (32). We cannot comment further for lack of details on steroid treatment, but the cumulative dose of steroid may be worth recording (33). Indeed, it is possible that the cumulative dose received was over the current recommendations for dexamethasone in ventilated COVID-19 patients not available at the beginning of the first wave (10, 12).

The second most frequent fungal disease was fungemia. The median onset was on the 16th day after ICU hospitalization. The global mortality rate was 45.7%, which was not higher than the 50.8% (690/1,358) observed in French ICUs (34). *C. albicans* was the main species, as reported elsewhere (3), the second one was *C. parapsilosis*, rather than *Candida glabrata*, as usually observed in the French adult ICU (34, 35). The patients who died were significantly older than those who survived, in contrast to CAPA. Host factors and steroid therapy during ICU stay had no impact on outcome of CA-fungemia. The mortality rate was higher if fungemia occurred sooner rather than later after ICU admission. This unexpected observation could be due to the late occurrence of *C. parapsilosis* fungemia, known to be associated with a better prognosis (34). Of note is the observation of environmental species such as *Clavispora lusitaniae* (4.9%), usually considered a rare species (<2%) associated with a low mortality (34). Additionally, the median stay in the ICU was 33 days (IQR, 18.35 to 51.75 days), longer than the mean stay in the ICU here for CAPA (median, 26 days), and as reported elsewhere (mean, 24.33 ± 18.88 days) for COVID-19 patients (12). This prolonged stay increases the nosocomial risk of fungemia. Overall, CA-fungemia may result from difficult working conditions (3, 4), resulting in catheter-related infection (36), which is well known for *C. parapsilosis* (37).

Only 17 CA-PCPs were reported in the ICU, which may be related to lack of investigation when host factors for PCP were not present. The mortality rate was 29.4%, close to the overall mortality recorded in the control group of the RECOVERY Study (10), knowing that most of the patients received co-trimoxazole. CA-PCP was strikingly diagnosed at entry in the ICU. Since *P. jirovecii* thrives on pulmonary alveolar cells according to the patient's immune status (5), the subsequent structural pulmonary damage following COVID-19 could hamper the development of this fungus. Thus, if pulmonary damage caused by viral infections favors the development of fungal coinfections (1, 2, 27), this may not be the case for *P. jirovecii*. Alternatively, the multiplication of the fungus could be controlled by nonspecific antibiotics such as azithromycin, as this drug prevents colonization from developing into full-blown PCP in cystic fibrosis patients (38). A more systematic assessment of these patients is needed to validate these hypotheses (39).

The retrospective design of our study prevented comments on the prevalence of

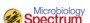

fungal infections in French COVID-19 patients. This prevalence varies between 0 and 33% (40) due to variations in case definitions and true variation between centers, but more importantly, variations in means and completeness of the investigations, with a complete workup for diagnosing fungal infection heavily dependent on clinical decisions. Other parameters not recorded, such as time and duration of mechanical ventilation, precise steroid prescription, or concomitant bacterial infections and antibiotic treatment, could have impacted the above results. The microbiological investigations were not standardized—hence, the difficulties in ascertaining the best way to diagnose fungal infections, including, for instance, the place of PCR in respiratory specimens. In addition, the EORTC/MSGERC imaging criteria (nodules, halos, cavities, wedge shaped, lobar or segmental consolidation, and tree in bud) were not recorded either. However, they are hardly analyzable in COVID-19 patients, although standardization is now proposed (41). Moreover, nodules with a halo sign are frequently seen in COVID-19 independently of invasive mold disease (42).

Our findings support more systematic screening for fungal markers in ICU patients with COVID-19 for better management of patients with coinfections, as already proposed (43). The three main fungal coinfections respond to different scenarios. CAPA occurred after 1 week of hospitalization in the ICU and could be hospital acquired. Its prognosis depended mainly on underlying diseases, with individuals with previous immunodepression or previous respiratory diseases having a worse prognosis. The more numerous microbiological criteria, the worse the prognosis. Thus, looking for additional microbiological documentation when the patient's condition deteriorates could be deleterious. Antifungal treatment initiation as soon as the first microbiological criterion is present may improve its efficacy (8). Moreover, since prescription of antimold treatment once the diagnosis was made did not seem beneficial, prophylaxis could be discussed for patients with the highest risk (18). CA-fungemia, on the contrary, occurred later and appeared not to be specifically related to COVID-19 but more likely to be linked to the difficult management of patients in ICUs. Episodes of CA-PCP were too few here to draw firm conclusions, but the time course suggested that *P. jirovecii* may play a role earlier in the course of COVID-19 before the patient enters the ICU.

## MATERIALS AND METHODS

**Data collection.** The NRCMA collaborating centers were asked for notification anonymously regarding all severe fungal diseases diagnosed in COVID-19 patients hospitalized in the ICU from 1 February to 31 May 2020, together with the means of diagnosis following local practices by skilled mycologists. The day of diagnosis was the day of the first positive microbiological test result transmitted to the clinicians as reported by the collaborators. In the absence of CAPA definitions at that time and given the challenge of imaging data analysis, microbiological evidence was considered for all mold infections: while a proven case required a biopsy (7), probable cases relied on a positive culture for mold, direct examination showing hyphae, and/or a positive *Aspergillus* PCR in respiratory specimens, bronchoalveolar lavage (BAL) fluid with a galactomannan (GM) index of >1, and/or in serum, a GM index of >0.5 or positive *Aspergillus* PCR. Molds were identified by morphology and/or matrix-assisted laser desorption ionization–time of flight mass spectrometry (MALDI-TOF MS) (44). Antifungal sensitivity testing was performed in each center for every isolate deemed responsible for invasive disease, and results were subsequently sent to the NRCMA for confirmation. GM was detected using Platelia *Aspergillus* Ag assay (Bio-Rad). Nucleic acid detection was performed locally using quantitative PCR (qPCR) with no restriction on the target (panfungal or germ specific), or the use of commercial or in-house assays, providing they comply with technical rules (45). Serum $(1{\rightarrow}3)$-$\beta$-D-glucan (BDG) was detected using the Fungitell kit (Associates of Cape Cod) with a positivity threshold of >80 pg/ml.

COVID-19-associated fungemia (CA-fungemia) was defined by the growth of yeasts from blood cultures. Species were identified by MALDI-TOF MS.

For the diagnosis of COVID-19 associated PCP (CA-PCP), commercial or in-house qPCR assays (46) were used with the quantitative cycle ($C_q$) value provided: the higher the $C_q$ value, the lower the fungal load. There was no restriction concerning the type of respiratory specimens tested.

The patients were grouped as follows: (i) according to the presence of immunosuppression factors following EORTC/MSGERC definitions (7), group 1; (ii) the presence of a history of respiratory disease (COPD or severe asthma), group 2; or (iii) neither of these two conditions, group 3. Only the first antifungal drugs administered at the time of diagnosis were recorded. Steroids prescribed for any underlying disease prior to COVID-19 (concerning potentially groups 1 and 2) were distinguished from steroids prescribed for COVID-19, knowing that no standard dose/regimen was then recommended. The patient's management was left to the clinician's discretion, with no impact of the present study on the clinician's decision.

**Statistical analysis.** When serial episodes of fungal infections were recorded for one patient (e.g., CA-fungemia followed by CAPA), only the first episode was considered for the global analysis to avoid autocorrelation. Subsequent analysis dealing specifically with CAPA, CA-fungemia, or CA-PCP included all episodes, whether the infection occurred first or second. Quantitative variables were described as means $\pm$ standard deviations or median and interquartile ranges ($Q_1$ to $Q_3$) according to normality of distributions. Univariate analysis was based on chi-square or Fisher's exact test for discrete variables. The Kruskal-Wallis test was used to compare medians for groups with unequal variances. Survival rates were determined by Kaplan-Meier analysis and compared by the log rank test. Data were analyzed using Stata software version 15.0 (Stata, College Station, TX).

**Ethics.** All data generated and interpreted were part of routine patient management, forming a prospective, consecutive cohort study covering the first 7 weeks of service, with 1-month follow-up, not requiring ethical approval. Data were recorded anonymously as part of the official duties of the NRCMA (Institut Pasteur Internal Review Board IRB 2009-34).

## SUPPLEMENTAL MATERIAL

Supplemental material is available online only.

**SUPPLEMENTAL FILE 1**, PDF file, 0.1 MB.

## ACKNOWLEDGMENTS

We thank Bertrand De Maupeou d'Ableiges for providing figures on mortality of Parisian ICUs. We thank the patients whose data were used and the many doctors and nonmedical staff who cared for the patients.

S.B. and A.A. designed the study. The manuscript was initially drafted by S.B., A.A., and F.D. S.B., K.S., and F.D. had full access to all the data in the study, verified the data, and did the statistical analysis. S.B. and A.A. had final responsibility for the decision to submit the manuscript for publication. All authors contributed to data acquisition, interpretation, and critical review and revision of the manuscript.

The NRCMA is supported by Santé Publique France and Institut Pasteur.

The following are members of the French Mycosis Study group: Julie Denis, Ferhat Meziani, Paul-Michel Mertes, Raoul Herbrecht, and Francis Schneider, Strasbourg; Julien Maizel, Rémy Nyga, Hervé Dupont, Céline Damiani, and Anne Totet, Amiens; Jean-Christophe Navellou, Laurence Millon, Emeline Scherer, and Gaël Piton, Besançon; Nicolas Argy, Sandrine Houze, Jean-François Timsit, Lila Bouadma, and Romain Sonneville, Hôpital Bichat, Paris; Jean Menotti, Damien Dupont, Charline Miossec, Jean-Christophe Richard, and Florent Wallet, Lyon; Marion Blaize, Julien Mayaux, and Charles-Edouard Luyt, Hôpital De La Pitié-Salpétrière, Paris; Emmanuel Dudoignon, Benoît Plaud, Elie Azoulay, Benjamin Chousterman, François Dépret, Blandine Denis, and Jean-Michel Molina, Hôpital Saint-Louis, Paris; Bruno Mégarbane, Sebastian Voicu, Magalie Collet, and Alexandre Mebazaa, Hôpital Lariboisière, Paris; Frédérique Boquel, Nicolas De Prost, and Keyvan Razazi, Hôpital Henri Mondor, Créteil; Fabienne Tamion, Benoit Veber, Marion Dehais, Gilles Gargala, and Loic Favennec, Rouen; Laurence Lachaud, Yvon Sterkers, Kada Klouche, and Romaric Larcher, Montpellier; Isabelle Accoceberry, Laurence Delhaes, Frédéric Gabriel, and Sébastien Imbert, Bordeaux; Louise Basmaciyan, Stéphane Valot, Eloïse Bailly, Denis Caillot, and Jean-Pierre Quenot, Dijon; Philippe Ichai, Paul Brousse; Stéphanie Ruiz, Benjamine Sarton, Stanislas Faguer, Emilie Guemas, Xavier Iriart, and Judith Fillaux, Toulouse; Nadia Anguel, Hôpital Du Kremlin-Bicètre, Villejuif; Julien Gautier, Antoine Goury, Maxime Hentzien, Yohan Nguyen, and Gaetan Deslée, Reims; Marie-Elisabeth Bougnoux Andremont and Fanny Lanternier, Hôpital Necker-Enfants Malades, Paris; Dorothée Quinio and Gilles Nevez, Brest; Victor Mercier, Claire Roger, Aurélien Daurat, Martin Mahul, and Cornelia Freitag, Nimes; Camille Courboulès, Faten El Sayed, and Valérie Sivadon-Tardy, Hôpital Ambroise Paré, Boulogne; Stéphane Gaudry, Marie Soulié, Nicolas Bonnet, and Yves Cohen, Hôpital Avicenne, Bobigny; Sophie Cayot and Philippe Poirier, Clermont-Ferrand; Kévin Brunet, Alida Minoza, and Arnaud W. Thille, Poitiers; Hélène Guegan and Florian Reizine, Rennes; Cyrille Chabartier, Elsa Sidani, Yohann Le Govic, and Jean-Marie Turmel, Fort-De-France, La Martinique; Marc Berthon, Nevers; and Anne-Marie Camin-Ravenne, Tarbes.

All data generated and interpreted was part of routine patient management, forming a prospective, consecutive cohort study covering the first seven weeks of service, with one-month follow-up.

We declare no conflicts of interest.

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
