## [Reviewer comments · Microbiology Spectrum]

Microbiology Spectrum

COVID-19 associated pulmonary aspergillosis, fungemia, and pneumocystosis in intensive care unit: a retrospective multicenter observational cohort during the first French pandemic wave.

Stephane Bretagne, Karine sitbon, Françoise Botterel, Sarah Dellière, Valérie Letscher-Bru, Taieb Chouaki, Anne-Pauline BELLANGER, Christine Bonnal, Arnaud Fekkar, Florence Persat, Damien Costa, nathalie Bourgeois, Frédéric Dalle, Florian Lussac-Sorton, André Paugam, Sophie Cassaing, lilia Hasseine, Antoine Huguenin, Nadia guennouni, Edith mazars, Solène Le Gal, Milène Sasso, Sophie Brun, Lucile CADOT, Carole Cassagne, Estelle Cateau, Jean-Pierre Gangneux, Maxime Moniot, Anne-Laure Roux, céline Tournus, Nicole Desbois-Nogard, Alain Le Coustumier, Olivier Moquet, Alexandre Alanio, and Françoise Dromer

Corresponding Author(s): Alexandre Alanio, Institut Pasteur

Review Timeline:

Submission Date:	August 3, 2021
Editorial Decision:	August 31, 2021
Revision Received:	September 16, 2021
Accepted:	September 17, 2021

Editor: Christina Cuomo

Reviewer(s): Disclosure of reviewer identity is with reference to reviewer comments included in decision letter(s). The following individuals involved in review of your submission have agreed to

reveal their identity: Jay Kolls (Reviewer #1)

Transaction Report:

DOI: <https://doi.org/10.1128/Spectrum.01138-21>

August 31, 2021

Prof. Alexandre Alanio
Institut Pasteur
Molecular Mycology Unit
28, rue du Dr Roux
Bat Duclaux aile Fourneau RDC Haut
Paris 75724
France

Re: Spectrum01138-21 (COVID-19 associated pulmonary aspergillosis, fungemia, and pneumocystosis in intensive care unit: a retrospective multicenter observational cohort during the first French pandemic wave.)

Dear Prof. Alexandre Alanio:

Thank you for submitting your manuscript to Microbiology Spectrum. Two reviewers have provided comments that I would like you to address in a revision.

When submitting the revised version of your paper, please provide (1) point-by-point responses to the issues raised by the reviewers as file type "Response to Reviewers," not in your cover letter, and (2) a PDF file that indicates the changes from the original submission (by highlighting or underlining the changes) as file type "Marked Up Manuscript - For Review Only". Please use this link to submit your revised manuscript - we strongly recommend that you submit your paper within the next 60 days or reach out to me. Detailed information on submitting your revised paper are below.

Link Not Available

Sincerely,

Christina Cuomo

Journals Department
Reviewer comments:

Reviewer #1 (Comments for the Author):

This is a useful report documenting the scope of fungal infections complicating SARS-CoV2 in France. However the data are presented in a haphazard way that I think would confuse the reader. Specific comments follow to try to improve the flow of the paper.

1. The authors focused on 244 subjects that required intensive care and then go on to present risk factors such as age, sex and mortality. However to be useful and to place these data in context it would be important to compare this group with COVID-19 without evidence of fungal infection. Is fungal infection an independent risk factor of mortality independent of age and sex?
2. The statistical test used to analyze the data in Figure 1 should be added to the Legend.
3. What was the denominator for the 283 fungal infections? What was the incidence of fungal infection?
4. Page 6: What are groups 1, 2, and 3? They are not defined at all in the results section. How were they chosen? Were they defined a priori, prior to the study or defined in a post-hoc manner?

Reviewer #2 (Comments for the Author):

This is a French multicentre retrospective study during 4 months (February 1st to May 31st 2020) reporting 3 encountered fungal co-infections in COVID-19 patients admitted to ICU. The authors present a large collection of enrolled patients and the study is of interest for mycologists and ICU doctors alike.

I have a few thoughts for authors consideration.

The authors suggest that CAPA could be hospital acquired because the median time to diagnosis was apparently 9 days. This can only be considered if screening of respiratory samples was done for all patients from day 0 onwards. Generally patients are already admitted with *Aspergillus* colonization, the latter is most common in patients with compromised airways for example due to COPD. But also smoking habits predisposes to *Aspergillus* colonization. The authors rightfully tone down the diagnosis of CAPA especially in the early months when no guidelines were available but the significance of the presence of *Aspergillus* in airway specimens (detected by culture, galactomannan antigen or specific PCR) remains to be fully understood. (I suggest to discuss this, also French study more in detail which was recently published in *Mycoses* 2021;64(9):980-988.). The authors discuss the occurrence of influenza associated aspergillosis to occur at a median of 3 days (line 262) and suggest herewith that this is community acquired. However influenza destroys the bronchial epithelium much more and quicker than SARS-CoV2 therefore the later diagnosis of CAPA may reflect this.

Secondly, I suppose that in the period feb-may very few patients may have been treated with systemic steroids (N Engl J Med. 2020; 10.1056/NEJMoa2021436). This may have impacted the diagnosis of CAPA (less due to less steroid use) in this first wave compared to the later waves. It is not clear how many patients received full doses of steroids. Line 317-318 states finding of a higher mortality in patients receiving steroids while line 320-321 states lack of details of steroid treatment. This may be biased since dexamethasone use in your study was not standard use for every patient as recommended by the Recovery trial. Sickest patients might have been selected in your study to receive "last resort" dexamethasone treatment. This influence may be discussed in more detail i.e. what was

proportion of steroids used in CAPA cases. There is only one (small) study comparing CAPA during first and second wave in Europe.

Thirdly, the authors state that "Aspergillus azole resistance was published only once" (line 204 and 316). Does this mean that all > 110 cultured Aspergillus was tested with AFST or azole plate screening to support this statement? If yes the results need to be mentioned (for example by stating the MIC-ranges and MIC90)

Line 315-316." A fumigatus azole resistance was mentioned only once?" or do you mean was "published only once (9)"? The latter is of course not correct since azole resistance and CAPA was also found once in Ireland (Med Mycol Case Rep. 2020; 10.1016/j.mmcr.2020.06.005)

and twice (2 cases out of 13 CAPA patients) in the Netherlands (Mycoses. 2021; 64(4): 457-464).

Line 361: does it also support routine AFST of patients with CAPA?

Line 364. CAPA could be hospital-acquired because it was diagnosed 1 week after ICU admission is speculation in the absence of systematic screening from day 0 onwards. See also above.

Minor suggestions:

In introduction: The study concerns CAPA however, only influenza associated aspergillosis is referenced (1,2). I don't see this relation especially since the latter disease is very different from CAPA. Therefore I suggest to discuss the first published (European) cases of CAPA in May 2020 (Mycoses.2020;63(6):528-534).

Line 136: "reported"you mean associated with COVID?

Line 169: delay or time

Line 253. ...are close to the published.....

Line 278. Rephrase ...should rise less question.....

Line 293.....important that..... important since

Line 301 decedents?

Line 315. In contrast to other settings. This paper(30) is not about pulmonary mucormycosis but on rhino-cerebral mucormycosis. Exchange with a CAM pulmonary study.

References: many of them are missing volume and pages numbering

Suppl table 1. Patient 3, Fongemia

Staff Comments:

Preparing Revision Guidelines

For complete guidelines on revision requirements, please see the journal Submission and Review Process requirements at <https://journals.asm.org/journal/Spectrum/submission-review-process>.

Submissions of a paper that does not conform to Microbiology Spectrum guidelines will delay acceptance of your manuscript. "

Please return the manuscript within 60 days; if you cannot complete the modification within this time period, please contact me. If you do not wish to modify the manuscript and prefer to submit it to another journal, please notify me of your decision immediately so that the manuscript may be formally withdrawn from consideration by Microbiology Spectrum.

If you would like to submit an image for consideration as the Featured Image for an issue, please contact Spectrum staff.

Reviewer comments:

Reviewer #1 (Comments for the Author):

This is a useful report documenting the scope of fungal infections complicating SARS-CoV2 in France. However the data are presented in a haphazard way that I think would confuse the reader. Specific comments follow to try to improve the flow of the paper.

1. The authors focused on 244 subjects that required intensive care and then go on to present risk factors such as age, sex and mortality. However to be useful and to place these data in context it would be important to compare this group with COVID-19 without evidence of fungal infection. Is fungal infection an independent risk factor of mortality independent of age and sex?

A: As explicitly explained in the manuscript (lines 350 to 362), the present study is a multicenter declarative study, and not a multicenter prospective longitudinal study with scheduled testing / imaging at entry and during hospitalization in all patients to decipher the risk factors for mortality in COVID-19 patients. This later study design was not possible in the emergency context of the first wave of COVID-19. Most ICU patients during the first wave were not tested for IFDs.

To compare as much as possible with the overall mortality, we obtain the overall mortality in some of the corresponding centers during the same period. This mortality was much lower than the mortality observed in our patients (Line 254: 22.6% vs 50.6% vs 22.6%, $p < 10^{-8}$), except for *Pneumocystis* positive patients.

2. The statistical test used to analyze the data in Figure 1 should be added to the Legend.

A: Done

3. What was the denominator for the 283 fungal infections? What was the incidence of fungal infection?

A: See comment above. In the absence of scheduled collection of microbiological tests in ICU patients, it is not possible to calculate an incidence. It is clearly presented in the discussion, where we did not speak about incidence (lines 350-362).

4. Page 6: What are groups 1, 2, and 3? They are not defined at all in the results section. How were they chosen? Were they defined a priori, prior to the study or defined in a post-hoc manner?

A: The three groups have been clearly defined in the Methods section, which unfortunately comes after the Results. We have put the information in the results (lines 185-187) to avoid any questions. Risk factors were included in the questionnaire sent to each participant prior to the analysis which intended to compare the three groups.

Reviewer #2 (Comments for the Author):

This is a French multicentre retrospective study during 4 months (February 1st to May 31st 2020) reporting 3 encountered fungal co-infections in COVID-19 patients admitted to ICU. The authors present a large collection of enrolled patients and the study is of interest for mycologists and ICU doctors alike.

I have a few thoughts for authors consideration.

1/ The authors suggest that CAPA could be hospital acquired because the median time to diagnosis was apparently 9 days. This can only be considered if screening of respiratory

samples was done for all patients from day 0 onwards. Generally patients are already admitted with *Aspergillus* colonization, the latter is most common in patients with compromised airways for example due to COPD. But also smoking habits predisposes to *Aspergillus* colonization.

A: We agree that screening at entry might help diagnose the timing of CAPA acquisition, but it would be difficult to perform BAL or even tracheal aspiration before these procedures are needed, except in one controlled study with ethical approval.

We cannot affirm from the literature and in our experience that all the patients had airway colonization. Indeed we think that is not expected in patients without a respiratory history (60% of the present CAPAs, Table 1). Of course, patients could be colonized between onset of COVID-19 and intensive care hospitalization, but this is highly speculative and there is no easy tool to determine this putative colonization. On the other hand, discussing the hospital acquisition as suspected by time of occurrence after admission (9 days) allows one to suggest preventive measures. For example, there is no mention in the literature that the air in intensive care units could be controlled for the fungal contamination of the environment or that basic measures to avoid *Aspergillus* contamination should be implemented and controlled, which could be discussed with our data (see lines 265-266 ref # 16).

The authors rightfully tone down the diagnosis of CAPA especially in the early months when no guidelines were available but the significance of the presence of *Aspergillus* in airway specimens (detected by culture, galactomannan antigen or specific PCR) remains to be fully understood. (I suggest to discuss this, also French study more in detail which was recently published in *Mycoses* 2021;64(9):980-988.).

A: We agree that the diagnosis of CAPA is not straightforward and that the definitions may help report epidemiological data. However, the comment we introduced in our discussion is that the more markers you have, the worse the prognosis. Therefore, waiting to be sure of the diagnosis could delay the treatment decision. Hence our comment (lines 292-293) to rely on culture, the main initial finding that is frequent as already done in ICU (ref # 23-24). Of course, understanding the meaning of a positive culture is mandatory, even though this question has been pending for many years.

The authors discuss the occurrence of influenza associated aspergillosis to occur at a median of 3 days (line 262) and suggest herewith that this is community acquired. However influenza destroys the bronchial epithelium much more and quicker than SARS-CoV2 therefore the later diagnosis of CAPA may reflect this.

A: We agree that the pathophysiology of IAPA and CAPA may be different, but the diagnosis of CAPA is usually made when the patient has already been hospitalized for a week. Our study simply shows the likelihood of CAPA being acquired in hospital, and this can lead to preventive measures (see answer above).

2/ Secondly, I suppose that in the period feb-may very few patients may have been treated with systemic steroids (N Engl J Med. 2020; 10.1056/NEJMoa2021436). This may have impacted the diagnosis of CAPA (less due to less steroid use) in this first wave compared to the later waves. It is not clear how many patients received full doses of steroids. Line 317-318 states finding of a higher mortality in patients receiving steroids while line 320-321 states lack of details of steroid treatment. This may be biased since dexamethasone use in your study was not standard use for every patient as recommended by the RECOVERY trial. Sickest patients might have been selected in your study to receive "last resort" dexamethasone treatment. This influence may be discussed in more detail i.e. what was proportion of steroids used in CAPA cases. There is only one (small) study comparing CAPA during first and second wave in Europe.

A: Contrary to the assumption that few patients received steroids in the first wave, to the question "Did the patient receive steroids for COVID-19" (to be differentiate from steroids for underlying disease) the response was positive for 49 of 146 responses (34%), which is not negligible. They also received higher doses than that recommended by the RECOVERY study in our experience (Delliere et al. CMI 2020). We have already written in our discussion that the cumulative dose is important and we have modified the paragraph in our discussion (lines 318-323) to make it clearer. As pointed out above, since this is a declarative study, to know the impact of steroids, it is necessary to know the steroids prescribed to all intensive care patients, which was beyond the scope of our study.

3/ Thirdly, the authors state that "Aspergillus azole resistance was published only once" (line 204 and 316). Does this mean that all > 110 cultured Aspergillus was tested with AFST or azole plate screening to support this statement? If yes the results need to be mentioned (for example by stating the MIC-ranges and MIC90)

A: All participating centers are mycology laboratories and all performed susceptibility testing using Etest when dealing with an invasive fungal disease and send the isolate for EUCAST confirmation to the NRCMA when azole resistance is suspected (vori MIC XXXX valeur CNR please). We cannot be sure that all isolates were tested, but we are sure that the majority was tested and had MICs < 1mg/L for ITC and VOR and < 0.25 for POS. MIC90 of French isolates are available on (<https://www.pasteur.fr/fr/sante-publique/CNR/les-cnr/mycoses-invasives-antifongiques->).

We have added (lines 390-391) to precise that AFST was performed in each center:

"Antifungal sensitivity testing was performed in each center for every isolate deemed responsible for invasive disease and subsequently sent to the NRCMA for confirmation if MICs were above EUCAST breakpoints."

Line 315-316." A fumigatus azole resistance was mentioned only once?" or do you mean was "published only once (9)"? The latter is of course not correct since azole resistance and CAPA was also found once in Ireland (Med Mycol Case Rep. 2020; 10.1016/j.mmcr.2020.06.005)

and twice (2 cases out of 13 CAPA patients) in the Netherlands (Mycoses. 2021; 64(4): 457-464).

A: Sorry for this misunderstanding. This was for the present study only. Changed for (line 317): "... mentioned only once in the present study and published separately (9)."

Line 361: does it also support routine AFST of patients with CAPA?

A: See answer above

Line 364. CAPA could be hospital-acquired because it was diagnosed 1 week after ICU admission is speculation in the absence of systematic screening from day 0 onwards. See also above.

A: See answers above.

Minor suggestions:

In introduction: The study concerns CAPA however, only influenza associated aspergillosis is referenced (1,2). I don't see this relation especially since the latter disease is very different from CAPA. Therefore I suggest to discuss the first published (European) cases of CAPA in May 2020 (Mycoses.2020;63(6):528-534).

A: We start recording in real time during the first wave, before Koehler's publication in Mycoses, which was therefore not our rationale to start our study. Instead, it is because aspergillosis complicates other viral pulmonary diseases that we start, hence the references quoted in our introduction. Moreover, we also start fungemia and PCP recording, which was not in Koehler's publication. However, this reference has been added in the discussion for the usefulness of performing tests for fungal infections.

Line 136: "reported"you mean associated with COVID?

A: Changed for also reported associated with COVID-19 (line 135)

Line 169: delay or time

A: Changed for "time" (line168).

Line 253. ...are close to the published.....

A: correction done (line 255).

Line 278. Rephrase ...should rise less question.....

A: Sentence changed as follow (line 279): "Antifungal prophylaxis could be proposed to patients with EORTC/MSGERC host factors and COVID-19 as after allogeneic stem cell transplantation or intensive chemotherapy (21).

Line 293.....important that..... important since

A: Changed for: Respiratory specimens are all the more important as the blood biomarkers are less efficient (line 294).

Line 301 decedents?

A: US dictionary: decedents = deceased person. It is written as such in ref#28. Unchanged

Line 315. In contrast to other settings. This paper (30) is not about pulmonary mucormycosis but on rhino-cerebral mucormycosis. Exchange with a CAM pulmonary study.

A: We have added a reference on pulmonary mucormycoses underlying the possible deleterious effect of steroids (ref#31).

References: many of them are missing volume and pages numbering

Suppl table 1. Patient 3, Fongemia

A: Done

September 17, 2021

Prof. Alexandre Alanio
Institut Pasteur
Molecular Mycology Unit
28, rue du Dr Roux
Bat Duclaux aile Fourneau RDC Haut
Paris 75724
France

Re: Spectrum01138-21R1 (COVID-19 associated pulmonary aspergillosis, fungemia, and pneumocystosis in intensive care unit: a retrospective multicenter observational cohort during the first French pandemic wave.)

Dear Prof. Alexandre Alanio:

Your manuscript has been accepted, and I am forwarding it to the ASM Journals Department for publication. You will be notified when your proofs are ready to be viewed.

Sincerely,

Christina Cuomo
Editor, Microbiology Spectrum
